# Responses of ticks to immersion in hot bathing water: Effect of surface type, water temperature, and soap on tick motor control

**David J. Schimpf[1], Matthew M. Ewert[2], Victor K. Lai[2]*, Benjamin L. Clarke[3]**

**1** Department of Biology, University of Minnesota – Duluth, Duluth, Minnesota, United States of America,
**2** Department of Chemical Engineering, University of Minnesota – Duluth, Duluth, Minnesota, United States of America, **3** Department of Biomedical Sciences, University of Minnesota – Duluth, Duluth, Minnesota, United States of America

* laix0066@d.umn.edu

**Data Availability Statement:** All relevant data are within the manuscript and its Supporting information files.

## Abstract

Preventing bites from undetected ticks through bathing practices would benefit public health, but the effects of these practices have been researched minimally. We immersed nymphal and adult hard ticks of species common in the eastern United States in tap water, using temperatures and durations that are realistic for human hot bathing. The effect of (a) different skin-equivalent surfaces (silicone and pig skin), and (b) water temperature was tested on *Amblyomma americanum*, *Dermacentor variabilis* and *Ixodes scapularis* nymphs. Overall, the type of surface had a much larger effect on the nymphs' tendency to stay in contact with the surface than water temperature did. Most nymphs that separated from the surface did so within the first 10 s of immersion, with the majority losing contact due to the formation of an air bubble between their ventral side and the test surface. In addition, adult *Ixodes scapularis* were tested for the effect of immersion time, temperature, and soap on tick responsiveness. Some individual adults moved abnormally or stopped moving as a result of longer or hotter immersion, but soap had little effect on responsiveness. Taken together, our results suggest that the surface plays a role in ticks' tendency to stay in contact; the use of different bath additives warrants further research. While water temperature did not have a significant short-term effect on tick separation, ticks that have not attached by their mouth parts may be rendered unresponsive and eventually lose contact with a person's skin in a hot bath. It should be noted that our research did not consider potential temperature effects on the pathogens themselves, as previous research suggests that some tickborne pathogens may become less hazardous even if the tick harboring them survives hot-water exposures and later bites the bather after remaining undetected.

## Introduction

With a growing burden of tickborne diseases in humans [1, 2], better prevention of tick-borne infections would be of great value to public health. Because known tick-vectored pathogens

**Funding:** The authors received no specific funding for this work.

**Competing interests:** The authors have declared that no competing interests exist.

are diverse and cause infections that often have overlapping or inconsistent signs or symptoms, timely diagnosis is more difficult; some of these diseases lack effective treatments [3]. Future emergence of new tickborne diseases would not be surprising. Additionally, tick bites (ticks attached by their mouthparts) sometimes induce paralysis [4] or serious allergic responses [5, 6], even without pathogen transmission. Prevention of tick bites avoids all of these potential problems. A reduced frequency of tick bites of humans may be accomplished by managing habitats to reduce tick populations and by increasing the effectiveness or wider adoption of preventive behaviors by individuals [7, 8]. It has recently been argued that habitat management to reduce tick contact with humans in the United States will need to expand to larger scales to be more effective [9, 10].

Evidence-based preventive personal behavior, however, has seen little recent research progress [11]. Mead [12] asserted that gaining a better understanding of the effect of bathing on the prevention of tick attachment would be one worthwhile venture. Population surveys for inferring the effectiveness of self-inspection and bathing have had mixed findings [13, 14]. Inspection of the body for ticks is facilitated by the disrobing that precedes bathing or showering, but what happens to ticks on skin that are not found by such inspections? Water pressure from a shower was said to be ineffective in removing attached ticks [15], although experimental evidence was not provided. The greater size of an adult tick makes it more likely to be detected on the skin than a nymph; however, nymphs vector many human infections [16] and can incite the other types of reactions to their bite. In this study, we investigated the response of adult and nymphal ticks to simulations of immersion in hot bath water to infer whether several physical factors of bathing water could induce non-attached ticks to lose contact with skin.

## Materials and methods

We tested flat nymphal *Amblyomma americanum* (lone star tick) and *Dermacentor variabilis* (American dog tick), as well as flat nymphal and adult *Ixodes scapularis* (black-legged tick), all members of the family Ixodidae (hard ticks). These were obtained from BEI Resources, Atlanta, which had cultured them on animals that were found to be free from pathogens known to be vectored to humans. No parasitic mites (family Acaridae) were seen on the ticks. All ticks were maintained in a closed container in air equilibrated with a saturated aqueous solution of $KH_2PO_4$, for a relative humidity of about 96% at the 19–22˚C holding temperatures [17]. The photoperiod was 14 h.

The test water was municipal service from a hot water tap. The pH ranged from 7.8–8.2 (Accumet Basic 15, Thermo Fisher Scientific, Waltham, MA) and the electrical conductivity ranged from 160–270 µS/cm (Hanna Instruments 98311, Woonsocket, RI) for both tap water and tap water plus soap (Kirk's Original Coco Castile). Water-air surface tensions at 40˚C were $67 \pm 4$ mN/m ($\pm$ SD, $n = 10$) for tap water, $64 \pm 6$ mN/m for tap water exposed to pig skin, or $40 \pm 1$ mN/m for tap water plus soap, all measured by the capillary rise method [18] using ImageJ 153a [19].

### Experiments with nymphs

We studied the effect of (i) the type of surface, and (ii) water temperature on nymphs' ability to stay in contact with the surface. The three species of nymphs were immersed on a 5×10×0.15 cm acrylic platform held at an angle of 45 degrees above horizontal by a steel frame (Fig 1). The upper surface of the platform was covered with either (a) a 5×10×0.1 cm slab of a proprietary silicone rubber-like material, or (b) raw pig skin from a farmer (A. Kozumplik) who raised and butchered the pig in Minnesota, both of which were kept taut across the platform by enameled-steel spring clamps. The silicone slab was 3-D printed using a Stratasys J750

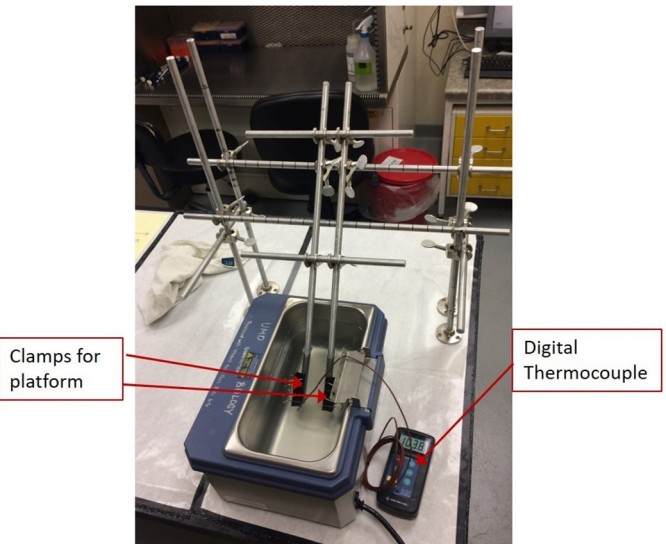

**Fig 1. Experimental test setup of the steel frame in the 5L water bath.** Real-time temperature is observed using the digital thermocouple. The adjustable rig is set at a platform angle of 45˚ for these experiments.

PolyJet 3D Printer (Stratasys Ltd, Rehovot, Israel); the material used had a reported Shore A hardness between 35–40, similar to that of other human skin surrogates [20]. Before testing, the hairs on the pig skin were removed with a straight-edge razor, and the skin was maintained in 1× Dulbecco's phosphate-buffered saline (Corning Inc., Corning, NY) at 5˚C. To test the effect of different platform surfaces, the testing apparatus was immersed in the thermostatic 5 L non-stirred water bath at two temperatures: (1) at 40.0 ± 0.3˚C (104˚F), which is the upper limit for approved hot tubs and spas (UL standard 1563), and (2) 44.5 ± 0.3˚C (112˚F), an initial bath temperature marginally tolerable for some persons. To further study the effect of water temperature, the silicone surface was tested at the intermediate temperatures 42.2 ± 0.3˚C and 43.3 ± 0.3˚C. The temperature was observed with a thermocouple meter (Cole-Parmer 91210–45, Vernon Hills, IL) via a 24-gauge insulated type-K sensor held near the middle of the water mass. The apparatus was removed from the bath, then the test surface was blotted dry with a cotton cloth and allowed to cool in air until it was at or just below 33.0˚C, as measured with a FLIR C2 Thermal Imaging Camera (FLIR Systems Inc., Wilsonville, OR). This made the test surface's temperature similar to that of normal human skin at the time that each nymph was placed on it. A single nymph was transferred to the middle of the test surface with an artist's brush; once we observed it to have good motor control, the apparatus was slowly immersed until the whole platform was underwater. Some ticks failed to show enough motor control before immersion and were discarded. We observed each nymph's behavior for 3 min after immersion or until it irreversibly lost contact with the platform, recording the elapsed time until contact was lost. Each nymph was used for a single test, with 20 nymphs of each species tested. Statistical summaries and hypothesis tests regarding dislodgement differences between species, temperatures, and surfaces were performed with *Statistix 9* [21].

## Experiments with adults

We studied the effect of immersion time and temperature on tick responsiveness in the presence and absence of soap. Equal numbers of male and female adult *Ixodes scapularis* were

tested for each time-temperature-liquid combination, with each individual used once. In addition to the plain tap water, half of the tests were performed with tap water containing soap. There is a lack of published estimates of soap quantities used in a bath [22]; we found the solid soap to lose about 40 mg (air-dry)/L by bathing with it and used that concentration [dissolved + micellar] in our tests. Adults were first acclimated for 5 h at 33˚C in humid air within capped 1.5 ml polypropylene centrifuge tubes in a thermostatic heating block, with temperature monitored in an adjoining tube by the same probe and meter. Acclimated individuals showing normal leg movement were then immersed in the heated test liquid in the same kind of tube, set in a different heating block. We monitored liquid temperature in an adjoining capped reference tube with the same probe and meter. The reference fluctuated by as much as 1–2˚C during an exposure. Adults were exposed to mean water temperatures in the range 40–45˚C for 15, 20, or 30 min, removed from the liquid, and observed for leg movement after being mechanically prodded periodically at room temperature while held in humid air.

## Results

### Experiments with nymphs

**Qualitative observation of nymph movement.** Upon immersion, every nymph walked from its initial position. There was no consistency in the direction of travel. Many walked visibly faster soon after immersion began. Some walked onto a clamp. A few remained motionless for extended intervals. Many nymphs tumbled down the inclined test surface, with some regaining control before losing complete contact and falling off the edge of the surface to the bottom of the bath. Nymphs that lost contact and sank to the bottom of the bath showed reduced movement there; they were all able to regain normal movement soon after they were removed from the water. However, most of the nymphs that lost contact floated off the platform, with the majority of the floaters lifted by a visible bubble between their ventral side and the test surface. In rare instances, there was a bubble visible between the animal and the test surface but the nymph did not lose contact, or a nymph separated itself from a visible bubble while retaining control on the test surface. Some of the floating ticks moved slowly to the wall of the bath, evidently in passive response to water surface forces rather than by active control. Once at the wall, floating nymphs were observed to be able to escape the water and climb all the way up the few cm of vertical brushed-stainless steel. A few nymphs floated when we dislodged them after 3 min of contact with the test surface.

**Effect of platform surface.** Many nymphs stayed on the test surface for 3 min. A significantly smaller proportion of the nymphs placed on pig skin lost contact within 3 min, as compared with those placed on silicone (Table 1); this was true at both temperatures. Apparent temperature effects were not consistent. Significantly more *Ixodes* lost contact at 44.5˚C than at

**Table 1. Number of nymphal ticks losing contact with platform within 3 minutes of immersion, out of 20 individuals tested for each combination of species, test surface, and temperature.**

| Species | Surface | | | |
|---|---|---|---|---|
| | Silicone | | Pig skin | |
| | 40˚C | 44.5˚C | 40˚C | 44.5˚C |
| *Amblyomma americanum* | 14[a] | 15[a] | 5[b] | 4[b] |
| *Dermacentor variabilis* | 18[a] | 18[a] | 8[b] | 0[c] |
| *Ixodes scapularis* | 12[a] | 19[b] | 5[ac] | 4[c] |

Numbers in the same line having the same letter in their superscript did not differ at $P < 0.05$ by two-tailed Fisher's exact test. No results in the same column differed at $P < 0.05$ by two-tailed Fisher's exact test.

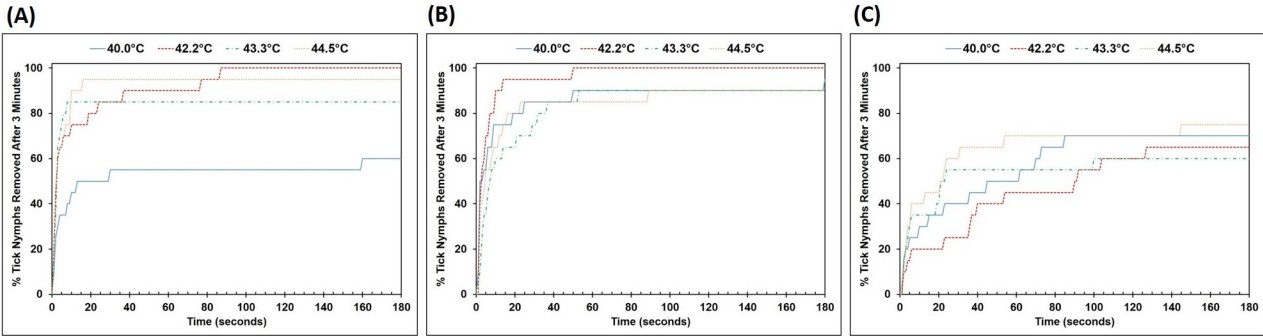

**Fig 2. Cumulative frequency graphs of percentage of (A)** *Ixodes*, **(B)** *Dermacentor*, **and (C)** *Amblyomma* **tick nymphs removed from the surface of the silicone skin equivalent with time.** Except for *Ixodes* nymphs at 40.0˚C which showed a lower removal percentage compared to the higher temperatures, there is no observable temperature dependence on tick nymph removal for all 3 species.

40˚C when on silicone, whereas the reverse was true for *Dermacentor* on pig skin. There were no significant differences among species in proportions of lost contact on the same surface at the same temperature (Table 1). When time until loss of contact was aggregated across all three species and both temperatures, the mean was 14 s on silicone, 39 s on pig skin (two-tailed *P* for normal approximation = 0.001, Wilcoxon rank-sum test). The median times were 4 s on silicone, 20 s on pig skin. These statistics exclude nymphs that never lost contact within 3 min.

**Effect of water temperature.** Fig 2 shows cumulative frequency graphs of the percentage of tick nymphs of each species separating from the platform within 3 minutes. In general, there was no observable temperature dependence, although a large difference in percentage of *Ixodes* nymphs separating was seen between 40.0˚C and the higher temperatures (Fig 2A). *Amblyomma* nymphs were able to stay on the silicone surface best, with < 80% of the nymphs losing contact after 3 minutes regardless of water temperature. For all species, a significant number of tick nymphs lost contact within the first 10 seconds upon immersion into the water bath; this phenomenon was less pronounced with the *Amblyomma* nymphs (Fig 2C). The majority of these tick nymphs that lost contact early on during the experiment floated off the surface due to the formation and subsequent detachment of an air bubble below them. This qualitative observation suggests that surface properties of the platform had a greater effect on nymphs retaining control than water temperature did.

### Experiments with adults

Among the adults tested, the female and male held in 41˚C plain water for 20 min did not lose leg movement by the end of that exposure (Table 2). All other ticks tested were unresponsive to prodding when first removed from their exposure tube. When eventual recovery of movement was observed, it was within 10–60 min (defined as "Soon" in Table 2). There was a tendency for adults exposed to higher temperatures or longer times to suffer lasting impairment of motor control, as inferred from lack of leg movement when prodding as late as the following morning (Table 2). This was seen in both plain water and soapy water.

## Discussion

### Applicability to human bathing

The tumbling by some nymphs that remained on the test surface suggests that more nymphs would have lost contact irreversibly had the angle of the platform been steeper. A seated bather

**Table 2. Responses of adult *Ixodes scapularis* after immersion in the specified conditions.**

| Liquid | Mean water temperature (˚C) | Time Exposed (min) | Recovery of Movement |
|--------|------------------------------|---------------------|----------------------|
| Water | 40 | 20 | Soon |
| Water | 41 | 20 | Immediate |
| Water | 43 | 20 | Soon |
| Water | 43 | 30 | Never |
| Water | 45 | 15 | Never |
| Soapy Water | 42 | 15 | Soon |
| Soapy Water | 42 | 20 | Soon |
| Soapy Water | 42 | 20 | Soon |
| Soapy Water | 43 | 20 | Partial by Next Morning |
| Soapy Water | 43 | 20 | Never |

Each line represents two ticks, one of each sex, both of which recovered at the same rate. "Soon" means full recovery within 10–60 min. "Never" means no recovery by the next morning.

would have a majority of their immersed skin surface steeper than 45˚. Movements by the person within the bath would change the angle for some locations on the skin. Ticks that lose contact may regain it if they are restrained by immersed hair or appressed skin folds (e.g., between toes or within the navel). Some nymphs walked upward on the platform, and a tick doing so on a bather may escape the water to a cooler non-immersed part of the body. Nymphs that float after losing contact could accidentally regain contact with the skin of a bather. This seems less likely for ticks that sink. The possibly lesser frequency at which nymphs floated off from pig skin hints that physical differences in the surface influence bubble development. A variety of dermatological care products, as well as hydrophobic foreign matter, may make human skin less like clean pig skin in this regard, but the importance of this is unknown. The possible formation of bubbles on the skin surface deserves exploration. We notice numerous small bubbles develop on hairs, fewer on skin, soon after our self-immersion in hot tap water. Bubbles and water from spa jets would probably tend to dislodge ticks from skin, and the filtering of recirculating spa water would reduce the chance of a tick regaining contact.

The difference between test surfaces in tick outcomes (Table 1) suggests that trials with live human subjects would be valuable. It would not be surprising if ticks on humans are less likely to lose contact under the same conditions than they were in these experiments. Voigt and Gorb [23] found that subaerial *Ixodes ricinus* did not maintain mechanical control as well on silicone as on live human skin. Further tests using silicone surrogates for skin should not be a priority, given those results and ours. The similar outcomes for each of the species of nymphs (Table 1) indicate that research with human participants may get generalizable findings without testing all three species. The strong tendency for adult ticks to stop movement when immersed for at least 15 min (Table 2) gives promise that a greater proportion of nymphs would lose contact when in hot water longer than the 3 min that we tested. Showering is a popular alternative to bathing, but its seemingly different physical properties call for specifically designed research on its preventive efficacy rather than extrapolation from immersion results.

## Possible enhancements to increase the effectiveness of bath water

A relevant question concerns whether safe substances in bath water could increase the proportion of ticks that separate from the skin. All of the ticks that we tested in soapy water stopped leg movement, at least temporarily (Table 2). A surfactant presumably reduces the tendency

for a layer of air to remain over a tick's spiracular plates during immersion [24], and perhaps some surfactant-charged water could penetrate the plate to reach more sensitive tissue; soap would not normally be in spa water. A concentration of soap or detergent slightly higher than what we used may be reasonable. In addition, surfactants could change surface tension properties and alter the likelihood of bubble formation to dislodge ticks, which may be worth further study. Some other commercial personal cleansing products may potentially be more distressing to ticks, and novel substances could also be researched for possible greater effectiveness as product ingredients or separate bath additives. These would not necessarily be overtly lethal compounds such as are used in treatments for human lice. For example, lone star ticks immersed at 19˚C had lower long-term survival in seawater (2% salinity) than in freshwater (0.2% salinity) [25]. Human skin can tolerate water much saltier than 2%, and if brief warm exposures to it proved highly effective at dislodging ticks, military installations with personnel at risk of tick bites might provide special facilities for controlled bathing or showering in it, perhaps with the use of detergent or potassium-based 'saltwater' soap that might increase liquid entry into the tick. It would also be worth knowing whether persons near the seashore may reduce the risk of tick bites by taking a dip in the ocean, which is somewhat >2% salinity at many beaches, soon after possible tick exposure.

## High temperatures and the infection process

While we looked at the effect of water temperature on ticks' ability to stay on a surface, another possible question for research is whether the brief exposure to higher temperatures would affect the efficacy of infection from ticks that are already attached or later attach after failing to separate from the skin. Nymphal black-legged ticks infected with *Borrelia burgdorferi* failed to infect mice when the ticks were first incubated at 33˚ or 37˚C for two weeks, whereas nymphs incubated at 15˚, 21˚, or 27˚C were able to infect mice [26]. It is unclear whether this result is due to an effect of temperature on the tick, or on the bacteria. There could be lasting effects of heat on pathogens within ticks that subsequently bite. For example, studies have shown that *B. burgdorferi* grew poorly in vitro above 41˚C [27], and in another experiment was permanently unable to grow when cooled after exposures ≥ 40˚C [28]; *Francisella tularensis* grew much more slowly in culture at 42˚ than at 35˚C [29]. Caution is warranted here, as it also seems possible that pathogens vectored from a tick that survived a hot bath could incite stronger autoimmune responses [30, 31]. These experiments imposed their higher temperatures for relatively long times; perhaps bath-duration exposures to ≈44˚C would have parallel effects. In another study, the tickborne eukaryotic pathogen of cattle *Babesia bigemina* lost substantial virulence after 15 min at 45˚C [32]; closely related organisms that induce babesiosis in humans might be similarly affected. Little seems to be known about how these hotter conditions affect pathogens within ticks, as opposed to those within culture media. Many tick-borne pathogens are obligately intracellular and have not been cultured above 37–40˚C. Even though the higher temperature that we used did not cause more nymphs to lose contact, it may be important for these other features of infection and disease severity. At supraoptimal temperatures, a few degrees more can cause large declines in organismal performance [33].

## Prospects, with limitations

Bathing or showering are widespread practices, including after outdoor activity that increases the risk of contact with ticks. Given the recent lack of new protective measures, this should be motivation for learning to what extent optimizing those practices could reduce the incidence of health problems from tick bites. Variables ripe for investigation include: time delay after first contact with a tick, duration of the bath or shower, water temperature, water motion, and

chemical composition of the bathing liquid. It would also be useful to know about effects on ticks that have begun attachment, effects on larvae, whether the pathogen load of a tick makes a difference (e.g., [34]), and effects on other tick taxa. Any eventual public suggestion that hot baths are an additional personal measure for preventing tickborne maladies should include a caveat that hot immersion is contraindicated for persons with certain conditions (e.g., [35, 36]). Many individuals who have not been taking hot baths should first seek clinical advice. Perhaps further research would find that some ticks lose skin contact at bath temperatures lower than those that we tested.

## Supporting information

**S1 Dataset.**
(XLSX)

## Acknowledgments

We thank Lyndon Ramrattan for help making the test apparatus, and the Medical Entomology Laboratory of the Centers for Disease Control and Prevention for providing the ticks.

## Author Contributions

**Conceptualization:** David J. Schimpf, Victor K. Lai, Benjamin L. Clarke.

**Data curation:** David J. Schimpf, Matthew M. Ewert, Victor K. Lai.

**Formal analysis:** David J. Schimpf, Matthew M. Ewert, Victor K. Lai.

**Investigation:** David J. Schimpf, Matthew M. Ewert, Victor K. Lai.

**Methodology:** David J. Schimpf, Matthew M. Ewert, Victor K. Lai.

**Project administration:** David J. Schimpf, Victor K. Lai.

**Resources:** David J. Schimpf, Victor K. Lai, Benjamin L. Clarke.

**Supervision:** David J. Schimpf, Victor K. Lai, Benjamin L. Clarke.

**Writing – original draft:** David J. Schimpf, Victor K. Lai.

**Writing – review & editing:** David J. Schimpf, Matthew M. Ewert, Victor K. Lai, Benjamin L. Clarke.

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
