## [Decision Letter · Decision Letter 0]

18 Oct 2021

PONE-D-21-27474Prospects for avoiding tick-borne illnesses through bathing: Effect of surface type, water temperature, and soap on tick attachmentPLOS ONE

Dear Dr. Lai,

Thank you for submitting your manuscript to PLOS ONE. After careful consideration, we feel that it has merit but does not fully meet PLOS ONE’s publication criteria as it currently stands. Therefore, we invite you to submit a revised version of the manuscript that addresses the points raised during the review process.

We look forward to receiving your revised manuscript.

Kind regards,

Maria Stefania Latrofa

Academic Editor

PLOS ONE

Journal Requirements:

2. In your manuscript, please provide the name of the butchery facility where you obtained the raw pig skin from.

3. Please amend your manuscript to include your abstract after the title page

Reviewers' comments:

Reviewer's Responses to Questions

**Comments to the Author**

1. Is the manuscript technically sound, and do the data support the conclusions?

Reviewer #1: Partly

Reviewer #2: Yes

Reviewer #3: Yes

2. Has the statistical analysis been performed appropriately and rigorously? 

Reviewer #1: No

Reviewer #2: Yes

Reviewer #3: Yes

3. Have the authors made all data underlying the findings in their manuscript fully available?

Reviewer #1: Yes

Reviewer #2: Yes

Reviewer #3: Yes

4. Is the manuscript presented in an intelligible fashion and written in standard English?

Reviewer #1: No

Reviewer #2: Yes

Reviewer #3: Yes

5. Review Comments to the Author

Reviewer #1: The authors are appreciated for the written article, however there are numerous aspects to consider. They present an highly ambitious title by mentioning "avoiding tick-borne illnesses", however, no tick-borne agent is evaluated throughout the work, much less the diseases they cause. In many sections, colloquial language, and the absence of accurate data from their results prevail, which is extremely negative in the case of an experiment. At the same time, the authors mention the importance of the surface in relation to the attachment and subsequent detachment of the ticks, which is why I consider it necessary to test their hypotheses on human subjects (as they mentioned in lines 183-184).

Reviewer #2: Dear authors

I reviewed the manuscript entitled “Prospects for avoiding tick-borne illnesses through bathing: Effect of surface type, water temperature, and soap on tick attachment” by Schimpf and colleagues. The manuscript reports important data about the biology of ticks and the prevention of tick borne pathogens to humans. Overall, the study was well planned and the text is well written, with novel aspects about these vectors. Therefore, the text deserves to be published on PLOS One as is it.

Reviewer #3: The article (PONE-D-21-27474), entitled “Prospects for avoiding tick-borne illnesses through bathing: Effect of surface type, water temperature, and soap on tick attachment”, provides potential future perspectives against the spread of tick-borne illness, discussing experimental data on tick attachment phenomena, according to different kind of environmental/external conditions. Overall, methods are technically sound (including laboratory procedures and statistical tools) and presented in a clear way, as well as results and their implications widely consistent. I do believe this manuscript deserves publication in PLOS ONE, pending minor revisions which are listed below.

Materials and methods

Line 49: dog tick Dermacentor variabilis is a kind of definition too general, considering other tick species which are recognized under this common name (e.g., Ixodes canisuga, Rhipicephalus sanguinesu sensu lato, etc…). Please, replace with “American dog tick”, which is the common name more largely employed to define Dermacentor variabilis.

Line 50: members of the family Ixodidae.

Line 52-53: “Ticks were tested after they had stopped excreting visible dark residue of blood ingested at their previous stage”. Here could be useful to report the timeline or at least an average of the number of days after which the tick specimens stopped to excreting visible dark residue of blood.

Line 54: family Acaridae.

Line 66: “acrylic platform held 45° above …”. Here the Authors have to clarify which degree (i.e., Celsius or Fahrenheit, etc…). Of course, they refer to °C but, anyway, reporting this information is absolutely necessary.

Line 95: Please, remove underline from “and temperature”.

Results

Table 2: In the caption of Table 2, the Authors provide a specific definition for “Soon” and Never, but quotation marks are reported only for soon, not for never. Please, add quotation marks also for the word never.

Line 226: Please, use the abbreviation B. burgdorferi, since you already define the scientific name in extenso at the line 221.

Line 238: Please, use features instead of facets.

6. PLOS authors have the option to publish the peer review history of their article (what does this mean?). If published, this will include your full peer review and any attached files.

Reviewer #1: No

Reviewer #2: No

Reviewer #3: **Yes: **Giovanni Sgroi

---

## [Author Response · Author response to Decision Letter 0]

19 Nov 2021

See attached Reviewer Response document.

---

## [Decision Letter · Decision Letter 1]

6 Dec 2021

Responses of ticks to immersion in hot bathing water: Effect of surface type, water temperature, and soap on tick motor control

PONE-D-21-27474R1

Dear Dr. Lai,

We’re pleased to inform you that your manuscript has been judged scientifically suitable for publication and will be formally accepted for publication once it meets all outstanding technical requirements.

Kind regards,

Maria Stefania Latrofa

Academic Editor

PLOS ONE

Reviewers' comments:

Reviewer's Responses to Questions

**Comments to the Author**

1. If the authors have adequately addressed your comments raised in a previous round of review and you feel that this manuscript is now acceptable for publication, you may indicate that here to bypass the “Comments to the Author” section, enter your conflict of interest statement in the “Confidential to Editor” section, and submit your "Accept" recommendation.

Reviewer #3: All comments have been addressed

2. Is the manuscript technically sound, and do the data support the conclusions?

Reviewer #3: Yes

3. Has the statistical analysis been performed appropriately and rigorously? 

Reviewer #3: Yes

4. Have the authors made all data underlying the findings in their manuscript fully available?

Reviewer #3: Yes

5. Is the manuscript presented in an intelligible fashion and written in standard English?

Reviewer #3: Yes

6. Review Comments to the Author

Reviewer #3: The Authors properly answered all points, improving the quality of the manuscript.

In its current version, I do believe this paper suitable for publication in PLoS ONE.

7. PLOS authors have the option to publish the peer review history of their article (what does this mean?). If published, this will include your full peer review and any attached files.

Reviewer #3: **Yes: **GIOVANNI SGROI

---

## [Editor Report · Acceptance letter]

9 Dec 2021

PONE-D-21-27474R1 

Responses of ticks to immersion in hot bathing water: Effect of surface type, water temperature, and soap on tick motor control 

Dear Dr. Lai:

I'm pleased to inform you that your manuscript has been deemed suitable for publication in PLOS ONE. Congratulations! Your manuscript is now with our production department. 

Kind regards, 

on behalf of

Dr. Maria Stefania Latrofa 

Academic Editor

PLOS ONE